# Determining Conditions for Thermoplastic Processing Guaranteeing Receipt of High-Quality Wire Rod for Cold Upsetting Using Numerical and Physical Modelling Methods

**DOI:** 10.3390/ma13030711

**Published:** 2020-02-05

**Authors:** Konrad Laber, Marcin Knapiński

**Affiliations:** Department of Plastic Processing and Safety Engineering, Faculty of Production Engineering and Materials Technology, Czestochowa University of Technology, 19 Armii Krajowej Ave., 42-200 Czestochowa, Poland; marcin.knapinski@pcz.pl

**Keywords:** wire rod with increased cold deformability, thermoplastic processing, numerical modelling, physical modelling, mechanical and technological properties

## Abstract

This paper presents the results of research with regard to determining the conditions of the thermoplastic processing of steel wire rod for cold upsetting, which ensures that a finished product with an even and fine-grained microstructure, without a clear banding and with increased cold deformability is obtained. The material used for the studies was 20MnB4 low carbon steel, and the studies were carried out on wire rod with a final diameter of 5.5 mm. Numerical modelling of the analysed process was carried out using commercial FORGE 2011^®^ and QTSteel^®^ programs, based on the finite element method. The GLEEBLE 3800^®^ metallurgical process simulator was used for the physical modelling studies. The obtained theoretical and experimental results were then verified in industrial conditions. Based on the obtained results, it was found that the optimum strip temperature before deformation in the RSM finishing block of the rolling mill is about 850 °C. The best cooling variant after the deformation process was the one in which the cooling rate was 10 °C/s. Such parameters of thermoplastic processing ensure that a final product with a favourable complex of mechanical and technological properties as well as a fine-grained, even microstructure, lacking clear banding, is obtained.

## 1. Introduction

Recently, huge progress has been made in rolling technology. In modern rolling mills, the final rolling rate reaches values up to 140 m/s, and the control systems used in the controlled cooling processes enable the treatment of rolled steel directly on the rolling line [1]. Many efforts have been directed at improving the efficiency and quality of manufactured products. The main challenges of the 21st century are energy saving and environmental protection. They require the modernization of the production technologies used to date or the development of new ones. Thermoplastic processing will be used for an increasing number of steel grades [1,2].

A significant problem during research related to the rolling of wire rod in modern rolling mills is the high dynamics of the process itself. Material deformation takes place in many passes (even over 30 passes) with a high strain rate, of the order of 2500 s^−1^, while the break times between final deformations are 0.05–0.02 s. For these reasons, precisely selected parameters of the deformation process and controlled cooling play an important role in shaping the microstructure and mechanical properties. These parameters must be adapted to the type of steel being processed, taking into account the required mechanical and technological properties of the finished product. Experimental studies in industrial conditions are expensive and usually do not enable the optimization of the process parameters. A rational way to significantly reduce the costs of modernization or implementation of new technologies is to use modern methods of numerical and physical modelling, combined with industrial verification [3].

Issues related to improving wire rod quality have been dealt with by the authors of papers [4,5,6,7,8,9,10,11,12,13]. In [5], several groups of steel for wire rod production were characterized, such as interstitial free, ferritic/martensitic and micro-alloyed pearlite steels. The possibilities of shaping the microstructure of these steels and the basic technological recommendations ensuring that a finished product with the desired properties is obtained were described. Paper [12] mainly concerns the possibility of improving the properties of low carbon steel wire rod by introducing niobium, boron and titanium into the steel, which favourably affect the mechanical properties of the finished product. The authors of this work showed that from the point of view of the area of application of the obtained wire rod, it is important to significantly increase the capacity for further, direct cold processing. Works [4,6,7,8,9,10,11,13] relate to the rolling processes of high carbon steel. In these works, the effect of the temperature and cooling conditions on the microstructure and properties of wire rod were analysed. On the other hand, works [14,15] presented a model of microstructure development during the rolling of C70D grade high carbon steel wire rod and the results obtained using it.

Works [16,17,18,19,20,21,22] concern issues related to heat exchange during controlled cooling of the wire rod on a roller conveyor. These works can be used during detailed studies of the cooling conditions of wire rod, e.g., to improve the uniformity of wire rod properties over its length. Work [18] presented a model to simulate wire rod cooling on a roller conveyor, taking into account all types of heat exchange as well as the packing density of the wire loops on the conveyor. The system for testing and recording the temperature of the wire rod during cooling on the roller conveyor was presented in [16]. It was used during industrial research, the aim of which was, among others, to determine the relationship between the cooling rate and the mechanical properties of the wire rod. In [22] a system for monitoring and controlling the properties of wire rod during cooling on the roller conveyor is presented, based on TTT (Time Temperature Transition) charts. 

In the available literature, no papers were found in which the authors explained in detail the impact of the strain rate during rolling of the wire rod, e.g., on the yield stress of the deformed material, development of the microstructure or the properties of the finished product. The impact of high strain rates on the yield stress of various steel grades was briefly described in [23,24,25,26,27,28,29,30].

Only two studies were found in the available literature [31,32] in which the problems of producing wire rod in rolling blocks were extensively described. According to the authors of [31], in the case of low carbon steels, the most advantageous properties are obtained by using slow cooling of the wire rod on the roller conveyor. As is clear from the research presented in this paper, cooling 20MnB4 steel at low rates promotes the formation of a ferritic–pearlitic banded microstructure, which reduces the technological properties of the finished product. Therefore, it is justified to accurately determine the cooling rate after the rolling process, which will ensure the receipt of a finished wire rod with the required complex of mechanical and technological properties, whose microstructure will be devoid of clear banding.

The authors of papers [33,34,35] were also involved in improving the quality of steel wire rod for cold upsetting. They stated that in the case of the studied steel, the best complex of mechanical properties of the finished product can be obtained after rolling in the final passes at 750 °C and subsequent cooling of the wire rod at a rate of 5 °C/s. The technological studies carried out made it possible to upset the studied steel with a relative plastic strain of 66% without losing the consistency of the material. However, no technological studies of the obtained wire rod were carried out in these works. The research results presented in this work are consistent with the results published in [33,34,35]. Nonetheless, as the research published in [3] showed, after the deformation of 20MnB4 steel for cold upsetting, at a rolling end temperature of about 750 °C, microstructure banding is obtained, which reduces the technological properties of the wire rod, regardless of the cooling rate used on the roller conveyor. For this reason, it is also justified to determine the correct temperature of the end of the rolling process, which, in combination with the appropriate cooling rate after rolling, will ensure that the finished product has the required microstructure and properties.

In the available technical literature, there are few papers on the rolling process of wire rod which describe the possibilities of shaping and improving the mechanical and technological properties of the finished product using numerical methods and physical modelling taking into account the limitations of the available testing apparatus and the verification of such studies in industrial conditions. Therefore, the research issues undertaken at work are current. An important achievement of the work is the solution of the numerical and physical modeling problems of the analyzed rolling process using commercially available software and test equipment, taking into account its limitations in terms of the applied total strain, strain rate and break times between successive deformations. The proposed methodology for modelling the rolling process of the wire rod reflects with high accuracy the actual technological process and the changes occurring in the microstructure of the deformed material. The proposed parameters of the thermo-plastic processing of wire rod from 20MnB4 steel grade with diameter of 5.5 mm ensure that a finished product with a microstructure and properties comparable with the products offered by leading world producers is obtained. The obtained results and their analysis should be helpful in developing changes in the currently used methods of wire rod production, or in the design of new technological lines for rolling wire rods.

## 2. Materials and Methods 

### 2.1. Materials

The research presented in the work was carried out on 20MnB4 low-carbon steel (Table 1) for cold upsetting, with the chemical composition according to PN-EN 10263-4:2004 [36].

In the case of wire rods intended for further cold working, an important parameter is the cold working capacity, determined in the upset test. In accordance with the applicable standards, such wire rods should have a minimum relative plastic strain of 50% and a sample height index after upsetting of 0.5 [37]. The wire rod for further cold forming produced by leading global manufacturers is characterized by a high level of mechanical and technological properties and the possibility of relative plastic strain of about 80% [38,39]. 

### 2.2. Process Characteristics

The research presented in the work was carried out for the entire production cycle of rolling the wire rod with a final diameter of 5.5 mm, for an example of a combined rolling mill (combination of bar rolling mill and wire rod rolling mill). The rolling process in the continuous rolling mill took place in 17 passes, while rolling in the wire rod rolling mill took place in 2 blocks: a 10-stand No-Twist Mill (NTM) block and a 4-stand Reducing Sizing Mill (RSM) block. According to the guidelines published in [31,40,41,42,43], in order to obtain a finished material with an even, fine-grained ferritic–pearlitic microstructure without clear banding, the final stage of deformation should take place in the austenitic range, when its temperature is about 30–80 °C higher than the temperature at the beginning of Ar_3_ austenite transformation. For the studied steel, the Ar_3_ temperature was 780 °C. In addition, according to work [31], in the case of low-carbon and low-alloy steels, intended for further cold forming, the most favourable temperature of loop arrangement is about 850–900 °C. This way of arranging the loops ensures that the increased plasticity of the metal is obtained, which is beneficial for the cold drawing process and makes it possible to shorten the recrystallizing annealing time after the drawing process [31]. Nevertheless, the temperature increase caused the deformation of the material in the RSM block, with high strain rates, which for 20MnB4 steel was about 50 °C, which should be taken into account here. Therefore, during the studies, the temperature of the rolled strip in the RSM block was 850 °C. Controlled cooling of the studied steel grade was applied after the rolling process. The heat treatment parameters are given in Table 2.

### 2.3. Numerical Modelling

The programs FORGE 2011^®^ (Transvalor, Sophia-Antipolis, France) and QTSteel^®^ (ITA-Technology and Software, MSL-Metaltech Services Ltd., Ostrava, Czech Republic) were used for the numerical modelling of the rolling process of the wire rod. The deformation parameters calculated in the FORGE 2011^®^ program were used to perform numerical modelling of microstructure development using the QTSteel^®^ program and to perform physical modelling using the GLEEBLE 3800 simulator (Dynamic Systems Inc. Poestenkill, NY, USA).

#### 2.3.1. Mathematical Model of FORGE 2011^®^ Program

In the FORGE 2011^®^ program, a mathematical model is used for the numerical modelling of the three-dimensional plastic flow of metal during rolling in grooves, in which the mechanical state of the deformed material is described using the Norton–Hoff law [44,45]:(1)$$S_{ij}=2K(T,{\dot{\varepsilon }}_{i},\varepsilon_{i}){\left(\sqrt{3}{\dot{\varepsilon }}_{i}\right)}^{m_{m}-1}{\dot{\varepsilon }}_{ij},$$
where *S_ij_*—strain tensor deviator, ${\dot{\varepsilon }}_{i}$—strain rate intensity, ${\dot{\varepsilon }}_{ij}$—strain rate tensor, $\varepsilon_{i}$—strain intensity, *T*—temperature, *K*—consistency depending on yield stress *σ_p_*, m_m_—coefficient characterizing hot metal deformation (0 < *m_m_* < 1).

The friction conditions prevailing on the contact surface of the material with the tools are described using the Coulomb and Treska friction models, in which the appropriate coefficient values are assumed [46]:(2)$$\tau_{j}=\mu \cdot \sigma_{n}\;\mathrm{for}\;\mu \cdot \sigma_{n}\leq \frac{\sigma_{p0}}{\sqrt{3}},$$
(3)$$\tau_{j}=m\frac{\sigma_{p0}}{\sqrt{3}}\;\mathrm{for}\;\mu \cdot \sigma_{n}>m\frac{\sigma_{p0}}{\sqrt{3}},$$
where *τ_j_*—unit friction force vector, *σ*_*p*0_—base yield stress, *σ_n_*—normal stress, *μ*—coefficient of friction, *m*—friction factor.

The temperature fields are calculated on the basis of differential equations describing temperature changes with transient heat flow [46]:(4)$$\frac{\partial }{\partial x}\left(k_{x}\frac{\partial T_{s}}{\partial x}\right)+\frac{\partial }{\partial y}\left(k_{y}\frac{\partial T_{s}}{\partial y}\right)+\frac{\partial }{\partial z}\left(k_{z}\frac{\partial T_{s}}{\partial z}\right)+\left(Q-c_{p}\rho \frac{\partial T_{s}}{\partial t}\right)=0,$$
where *k_x_*, *k_y_*, *k_z_*—distribution functions of anisotropic thermal conductivity coefficients in x, y, z directions, *T*_s_—function describing the temperature in the zone in question, *Q*—rate distribution function for generating strain heat, *c_p_*—specific heat distribution function of the deformed material, *ρ*—density distribution function. 

The boundary conditions were adopted as the combined boundary conditions of the second and third types, in the form [46]:(5)$$k_{x}\frac{\partial T_{s}}{\partial x}l_{x}+k_{y}\frac{\partial T_{s}}{\partial y}l_{y}+k_{z}\frac{\partial T_{s}}{\partial z}l_{z}+q+\alpha_{k}T_{s}=0,$$
where *l_x_*, *l_y_*, *l_z_*—directional cosines normal to the surface of the deformed strip, *q*—heat flow rate on the surface of the cooled zone, *α_k_*—convection losses. 

Equations (4) and (5) clearly determine the heat exchange during modelling of the rolling process. 

The input data for numerical modelling of the analysed rolling process are given in Table 3. These data were adopted on the basis of the technical literature and previous experience.

#### 2.3.2. Initial and Boundary Conditions of Wire Rod Rolling Process

The initial and boundary conditions necessary for numerical modelling of the analysed rolling process of the 5.5 mm wire rod are given in Table 3. These data were determined on the basis of the technical literature [43,47,48,49,50,51]. Other data necessary to carry out the numerical modelling of the rolling process (initial temperature, relative rolling reduction, strain times and break times, roller rotational rates and linear rate of the strip) were adopted on the basis of the industrial data. In the analysed process, the average temperature on the cross-section of the charge before the first stand of the rolling mill was 1130 °C and the average temperature of the side surface was 1075 °C. The thermophysical properties of 20MnB4 steel (Table 4) were taken from material database of FORGE 2011^®^ software.

#### 2.3.3. Rheological Properties of 20MnB4 Steel

The Hensel–Spittel equation was used to describe the rheological properties of the studied steel [46]:(6)$$\sigma_{p}=A\cdot e^{m_{1}\cdot T}\cdot T^{m_{9}}\cdot \varepsilon^{m_{2}}\cdot e^{\frac{m_{4}}{\varepsilon }}\cdot {\left(1+\varepsilon \right)}^{m_{5}\cdot T}\cdot e^{m_{7}\cdot \varepsilon }\cdot {\dot{\varepsilon }}^{m_{3}}\cdot {\dot{\varepsilon }}^{m_{8}\cdot T},$$
where *σ_p_*—yield stress, MPa, *T*—temperature, °C, ε—true strain, $\dot{\varepsilon }$—strain rate, s^−1^, A, *m*_1_–*m*_9_—coefficients.

During the numerical modelling of the rolling process of 5.5 mm wire rod in the NTM and RSM blocks of the rolling mill, the rheological properties of the studied steel were defined using Equation (6) and the coefficients given in Table 5.

When determining the rheological properties of the investigated steel for the rolling conditions in the wire rod rolling mill (NTM and RSM blocks), the results of published studies, among others, in works [31,32] were taken into account, in which the values of yield stress for steels with a similar chemical composition to 20MnB4 steel were determined, in terms of the strain rate and temperature occurring during the rolling of 5.5 mm diameter wire rods. By extrapolating the values of yield stress of the examined steel for the strain rate occurring during the rolling of the 5.5 mm diameter wire rod, values consistent with those published in papers [31,32] were obtained.

#### 2.3.4. Mathematical Model of QTSteel Program^®^

In the QTSteel^®^ program, when forecasting the microstructure and mechanical properties of heat treated or thermoplastically processed steel, data from the cooling curves on the TTT chart are used. Calculating the percentage content of the microstructure components is performed step by step for the relevant sections of the cooling curve. To describe the kinetics of the transformation of individual components of the microstructure, the program uses the Avrami equation (7) [52,53]:(7)$$X_{i}(T,t)=(1-\exp(-k(T)\cdot t^{n(T)}))\cdot X_{\gamma },$$
where: $X_{i}(T,t)$—volume fraction of individual components of the microstructure: ferrite, perlite, bainite, *k*(*T*) and *n*(*T*)—parameters depending on the transformation mechanism and places of privileged nucleation and on the cooling rate, calculated on the basis of TTT charts for a given temperature, *T*—temperature, *t*—time, $X_{\gamma }$—volume fraction of residual austenite.

The volume fraction of martensite during martensitic transformation is calculated on the basis of the Koistinen–Marburger equation [53]:(8)$$X_{m}(T)=(1-\exp(-b\cdot {\left(T_{ms}-T\right)}^{n}))\cdot X_{\gamma },$$
where: $X_{m}$—volume fraction of martensite, *b*, *n*—constant, $T_{ms}$—martensitic transformation start temperature, *T*—temperature, $X_{\gamma }$—volume fraction of residual austenite.

Vickers HV hardness is determined by means of a regression equation [52,53]:(9)$$HV=C_{0}+X_{f}\cdot \sum (D_{i}\cdot c_{i}+X_{p}\cdot \sum E_{i}\cdot c_{i}+X_{b}\cdot \sum F_{i}\cdot c_{i}+X_{m}\cdot \sum G_{i}\cdot c_{i},$$
where: *HV*—Vickers hardness, $X_{f}$, $X_{p}$, $X_{b}$, $X_{m}$—volume fractions: ferrite, perlite, bainite, martensite, *C_0_*, *D_i_*, *E_i_*, *F_i_*, *G_i_*—constant, *c_i_*—percentage of alloying additions.

The tensile strength was determined based on Equation (10) [52]:(10)UTS=f(HV)=−a+b·HV,
where: *UTS*—Ultimate tensile strength, *HV*—Vickers hardness, and *a*, *b*—constant.

Yield strength YS is determined by Equation (11) [52,53]:(11)$$YS=f(D_{\alpha },C_{r},X_{f},\sum \left(X_{p}+X_{b}+X_{m})\right),$$
where: *D_α_*—ferrite grain size, *C_r_*—cooling rate, $X_{f}$, $X_{p}$, $X_{b}$, $X_{m}$—volume fractions: ferrite, perlite, bainite, martensite.

Detailed results of research carried out using the DIL 805 A/D dilatometer [54], the aim of which was to determine the phase transition temperatures, develop TTT and DTTT (Deformation Time Temperature Transition) charts and to determine the most favourable cooling conditions for 20MnB4 steel, were published, among others, in [55]. Taking into account the obtained results, the DTTT graph (Figure 1) was used to determine the impact of the cooling conditions on the forming of the wire rod microstructure immediately after the deformation process. The characteristic temperatures of phase transitions and hardness of 20MnB4 steel are presented in Table 6.

It was found that in order to obtain a ferritic–pearlitic microstructure in the finished product, the cooling rate should not exceed 15 °C/s. Increasing the cooling rate above 15 °C/s causes the formation of bainite, bainitic-martensitic and martensitic structures in the material, which results in deterioration of the ability of the investigated steel for further cold working, or in extreme cases prevents it.

### 2.4. Physical Modelling

The currently used wire rolling technologies are characterized by high dynamics of the deformation processes [2,56]. This creates major problems when physically modelling these processes using available laboratory equipment. Strain parameters (ε, $\dot{\varepsilon }$, *T*) occurring in real technological processes affect the nature of changes in the yield stress of the deformed material, and thus the microstructure and properties of the finished product. 

Physical modelling of the rolling process of the wire rod was carried out in uniaxial compression studies using the GLEEBLE 3800^®^ metallurgical processes simulator, using cylindrical samples with diameter d = 10 mm and height h = 12 mm. The frontal surface of the samples and tool surfaces were separated by tantalum, graphite foil and a special graphite-based lubricant to minimize friction and increase uniformity of the deformation. Temperature control was carried out using K-type thermocouples (NiCr-NiAl).

During the physical modelling of the rolling process of a 20MnB4 steel wire rod with a diameter of 5.5 mm, the experience gained during the implementation of previous research related to the physical modelling of actual plastic forming processes was used [56,57,58]. The research published in papers [56,57,59] shows that in the analysed rolling process and for the studied steel, the development of the microstructure and mechanical properties of the finished product are significantly affected by the deformation conditions provided in the last four passes and the method of strip cooling during and after the rolling process. Therefore, the physical modelling of the rolling process of 5.5 mm wire rod made of 20 MnB4 steel was carried out for the deformation conditions occurring in the last few passes of the rolling process (RSM block of the rolling mill). In the analysed process, the break times between the last four deformations are less than 0.01 s. Accurate physical modelling of the four-deformation cycle while maintaining the appropriate break times between successive passes is impossible when using existing equipment. At the same time, with such small times, the microstructure of the investigated material should not change significantly. Therefore, during physical modelling of the rolling process, the deformation in the last four passes was replaced by one deformation with a value equal to the sum of the individual four deformations. This methodology is acceptable, which is confirmed by the results of published studies, among others, in work [58].

As the results from the data presented in Figure 2 show, the level and course of changes in yield stress are similar in both cases. Slight differences only appear at the end of the deformation. This may be due to the fact that in the last two deformations (curve No. 1) the achieved strain rate was about 50 s^−1^ and 28 s^−1^ and it was lower than the set values (213 s^−1^ and 141 s^−1^). The reason for this is the short break time between deformations, preventing the achievement of such high strain rates. Based on the obtained research results, it was found that replacing the sequence of the last four deformations with one deformation will not cause a large error in the analysed case [58]. 

Another important problem during the physical modelling of the rolling process is the high strain rate, which in the final stage of the analysed process exceeded 2000 s^−1^. Due to the inability to use such a high value of strain rate during the physical modelling of rolling with the GLEEBLE 3800^®^ simulator, a strain rate of about 250 s^−1^ was used, utilizing the research results published in [31], in which the authors conducted studies on the impact of high strain rates on yield stress, for various steel grades. These studies were conducted on a specially constructed device in which the deformation conditions were similar to those found in the actual rolling process of wire rod. Figure 3 shows the results of changes in yield stress for steels with a chemical composition similar to low-carbon steels for cold upsetting for various deformation conditions.

There is a certain strain rate limit (about 250 s^−1^) beyond which the yield stress does not show significant changes. This tendency occurred for all the steel grades examined in [31].

As shown in the results from the research published in [56], during physical modelling it is permissible to use the limit value of the strain rate above which the yield stress does not change. The results of metallographic studies and the analysed mechanical properties of the material after physical modelling correspond with high accuracy to the results obtained in industrial conditions.

During the physical modelling of the rolling process, the samples were heated at the rate of 7.5 °C/s to the temperature corresponding to the temperature of the strip after the 17th rolling stand of the continuous bar rolling mill and held at this temperature for 60 s. Then, the temperature change in time was programmed in such a way that after the time corresponding to the movement of the strip between rolling stand No.17 of the continuous rod rolling mill and the RSM block of the wire rod rolling mill, the required value of the material temperature was obtained. The sample was then held at this temperature for 5 s and then deformed. The total strain value of the samples corresponded to the total strain in the RSM block. After the deformation process, the samples were cooled in accordance with the parameters given in Table 2.

## 3. Results

### 3.1. Numerical Modelling Results

Figure 4 shows examples of temperature distributions, strain intensities, strain rate intensities and stress intensities obtained as a result of numerical modelling the rolling process of the 5.5 mm diameter wire rod from 20MnB4 steel using the FORGE 2011^®^ program.

Based on the data presented in Figure 4a, it was found that the average cross-sectional temperature after the rolling process in the first rolling stand of the RSM block was about 880°C, while the average lateral surface temperature was about 870 °C. The highest temperature values were observed in the central part of the rolled steel and the area with the lowest temperature was the side surface not in contact with the walls of the groove (about 845 °C). Analysing the data presented in Figure 4b–d, it was found that the distribution of strain intensity, strain rate intensity and stress intensity of the rolled strip in the first stand of the RSM block is characteristic for rolling in grooves. With regard to the strain intensity (Figure 4b), it was found that the greatest strain intensity was found in the central areas of the rolled strip, while the lowest strain intensity was observed in the free zones of rolled metal, not limited by the walls of the groove.

Figure 5 shows the temperature changes throughout the rolling cycle. Analysing the data presented in Figure 5, it was found that after the deformation process of the tested steel in rolling stand No. 1, the temperature of the cross-section of the rolled strip was about 1125 °C, while the average temperature of the side surface of the strip was about 1050 °C. During rolling in the continuous medium rolling mill, a decrease in the strip temperature value was found at the initial stage of the rolling process, in the box groove system when small elongation coefficients are used and the rolling rate is relatively low. Due to the long contact time of the strip with the rollers and the long break times between successive passes, strong cooling of the surface of the rolled strip occurred. 

In accordance with the principles of continuous rolling in subsequent passes, the rolling rate increased proportionally to the elongation of the rolled strip. The increase in rolling rate resulted in shortening of the break time and material cooling between further passes. In subsequent rolling stands, elongation grooves in the oval-circle system were used, in which the elongation coefficients and set strain were higher. This caused an increase in the plastic strain energy and an increase in the temperature of the deformed strip in subsequent passes. In addition, during the rolling process, the heat from areas located in the central part of the strip was conducted towards the surface. Based on the data presented in Figure 5, it can be stated that in subsequent passes the difference between the average temperature and surface temperature was reduced, which to some extent was caused by the decrease in the cross-section of the rolled strip. As a result of accelerated water cooling in the cooling zone before the NTM block of the rolling mill, the average cross-section temperature dropped to around 860 °C, while the average side surface temperature was around 850 °C. As a result of intensive deformation of the strip in the NTM block with a high strain rate, the temperature of the rolled steel increased. The average cross-sectional temperature of the 20MnB4 steel after the last stand of the NTM block was close to the average surface temperature and was about 1000 °C. As a result of intensive cooling of the strip with water in the cooling zone located in front of the RSM block, its temperature was reduced to about 860 °C. After rolling the 20MnB4 steel in the RSM block, the average cross-sectional temperature was close to the surface temperature and was about 900 °C. Immediately after the rolling process, the material was cooled in two stages in a controlled manner using air blowing (Table 2).

The numerically determined average values of strain intensity and strain rate intensity in the material throughout the entire production cycle are given in Table 7. This table also contains the maximum stress intensity values of the tested steel, which (for the RSM block) were compared with the values obtained during physical modelling of the rolling process with using the GLEEBLE 3800 simulator. The stress values of the tested material obtained during physical modelling are the maximum values (calculated on the basis of the maximum value of the compressive force).

The analysis of the changes in the average value of the strain rate intensity of the examined steel grade during rolling in a continuous rolling mill (passes No. 1–17) shows that it increased in subsequent rolling stands, which was mainly caused by the increase in rolling rate and the set rolling reductions. An exception occurred when rolling bars constituting the charge for rolling wire rod with a diameter of 5.5 mm in pass No. 17, in which a slight decrease in the mean value of the strain rate intensity in the strip was observed, compared to the values of this intensity in the rolled strip in pass No. 16.

This change was caused by a lower value of strain intensity. After analysing the distribution of the average value of the strain rate intensity of the tested material during the rolling of 5.5 mm wire rod in the NTM block (passes No. 18–27), it was found that its constant and dynamic growth occurred, mainly caused by the increase in the value of the rolling reductions and the increase in the linear rolling rate. Based on the analysis of the changes in the average value of strain rate intensity during rolling in the RSM block (passes No. 28–31), it was found that this intensity gradually decreased, which was caused by a smaller value of strain intensity in subsequent stands of this block. Based on the analysis of changes in the stress intensity value of 20MnB4 steel during rolling in the continuous rolling mill (passes No. 1–17), it was found that this intensity increased in subsequent passes. At the initial stage of the rolling process, this was due to a decrease in the temperature of the rolled steel, while in subsequent passes it was mainly due to an increase in the strain intensity of the tested material. Owing to the slight decrease in the intensity of strain and the intensity of strain rate in the rolled strip in stand No. 17 (compared to the values calculated for strips deformed in roll stand No. 16), a slight decrease in stress intensity was also found in this pass. By analysing the changes in the stress intensity in the strip during the rolling of 5.5 mm wire rod in the NTM block (passes No. 18–27), it was found that the values of this intensity alternately increased and decreased, which was caused by the employed groove system. In the NTM block, along with the rapid increase in the strain rate intensity in the strip, there was an increase in the stress intensity value, but at the same time a significant increase in the temperature of the rolled steel, whereby the impact of the strain rate intensity on the stress intensity in the deformed material was smaller. Analysis of the research results on the changes in the stress intensity in 20MnB4 steel during the rolling of 5.5 mm diameter wire rod in the RSM block (passes No. 28–31) shows that this intensity decreased its value in subsequent rolling stands. This was caused by a decrease in the strain intensity and the strain rate intensity of the studied steel grade and by an increase in temperature in the deformed strip, especially in the first two stands of the RSM block. Comparing the numerically determined maximum values of stress intensity of the investigated material (for the stands of the RSM block) with the values obtained during the physical modelling of the analysed rolling process (Chapter 3.2), they were found be highly convergent. On this basis, it was found that the rheological properties of the examined steel, friction conditions and heat transfer coefficients adopted for numerical modelling were correctly determined.

As a result of the numerical modelling of the microstructure development using the QTSteel^®^ program, the distribution of changes in the austenite grain size in individual passes (Table 8) was determined.

On the basis of preliminary studies, it was established that in the analysed process the initial grain size of 20MnB4 austenite was about 200 µm. The data in Table 8 shows that, during the rolling process, the average austenite grain size gradually decreased, reaching 44 µm in the last continuous mill stand (pass No. 17). In the first pass, the strain value was too low (0.18) to start recrystallization, and the austenite grain expanded as a result of the long break time after deformation. Analysing the data for the first 17 passes, a slight increase in the austenite grain size of the tested steel was observed in passes No. 3, 7, 11 and 13. This could be caused by a lack of recrystallization resulting from a too low strain value, which was less than the critical value necessary to start the softening processes, and in the case of exceeding this value, a too short break time between deformations. During the rolling of the studied steel grade in the first stand of the NTM block (pass No. 18), the austenite grain size was 43 µm. During the deformation of the investigated steel in subsequent stands of the NTM block (passes No. 18–26), the austenite grain size was in the range of 17–20 µm. In the last rolling stand of the NTM block, the austenite grain size was 42 µm. As a result of the deformation of the examined material in the first rolling stand of the RSM block (pass No. 28), the austenite grain size reached about 21 µm. During strip deformation in subsequent rolling stands of the RSM block, the austenite grain size decreased to 18 µm. 

The percentage share of individual components of the microstructure on the cross-section of 20MnB4 steel wire rod is shown in Figure 6. As results from the data presented in Figure 6 show, during the cooling of the wire rod immediately after the rolling process the percentages of individual phases changed from 88% ferrite and 12% perlite for the cooling rate of 0.6°C/s, to 81% ferrite and 19% perlite when cooling at 15°C/s. Based on the analysis of the results of numerical modelling of microstructure development, it was found that in the studied range of temperature and cooling rate the obtained wire had a ferritic–pearlitic structure. The percentage of individual components of the microstructure depended on the rate of controlled cooling after the rolling process. At the same time as the cooling rate increased, a gradual decrease in the ferrite percentage and an increase in the perlite percentage were observed.

Figure 7 presents the research results on the impact of the cooling rate on the change in selected mechanical properties of 20MnB4 steel obtained during numerical modelling using the QTSteel^®^ program.

Analysing the data presented in Figure 7, it was found that in the examined range of cooling rate after rolling, both the value of the offset yield strength of the tested steel as well as its tensile strength increased along with an increase in the cooling rate. After cooling 20MnB4 steel at the rate of 0.6 °C/s, the value of the offset yield strength was about 314 MPa, while the ultimate tensile strength was 512 MPa. However, after cooling the tested material at the rate of 15 °C/s, the values of the analysed parameters increased to 400 MPa in the case of offset yield strength and to 600 MPa in the case of tensile strength, respectively. Along with an increase in the cooling rate after rolling, a simultaneous increase in the plasticity reserve (YS / UTS) was also observed, which varied from 0.61 to 0.67.

### 3.2. Physical Modelling Results

Figure 8 shows the course of changes in yield stress during the physical modelling of the rolling process of the wire rod in the RSM block, using the GLEEBLE 3800^®^ simulator.

The analysis of the course of changes in the yield stress of the tested steel grade shows that there is a rapid increase in the initial stage of deformation (to a strain value of about 0.15), caused by a high strain rate of about 250 s^−1^. At a later stage of the deformation process (up to a strain value of about 0.8), the yield stress value increased slightly. It can be stated that the stress values obtained during the physical modelling of the 5.5 mm diameter rolling process in the RSM block are similar to the results obtained during numerical modelling (Chapter 3.1, Table 7). In the case of physical modelling of the rolling process of this wire rod, the largest error did not exceed 15%. This also indicates the correctly determined rheological properties of the examined steel used during numerical studies, and that at strain rates greater than 250 s^−1^ the yield stress of the studied steel does not change significantly. The oscillations of the yield stress of 20MnB4 steel observed in Figure 7 were not due to the material properties and were caused by the hydraulic system of the GLEEBLE 3800 simulator at the high strain rate, which was the limit value possible to obtain in the device.

After the physical modelling of the rolling process of the wire rod in the RSM block, samples were then made for metallographic studies and for testing selected mechanical properties. Selected mechanical properties were determined from dependences (12) and (13) [60], based on the chemical composition and the average size of ferrite grain of the investigated steel.
(12)$$YS=62.6+26(\%Mn)+60(\%Si)+759(\%P)+213(\%Cu)+3286(\%N)+\frac{19.7}{\sqrt{\frac{D_{\alpha }}{1000}}},$$
(13)$$UTS=165+54(\%Mn)+100(\%Si)+652(\%P)+473(\%Ni)+635(\%C)+2173(\%N)+\frac{11}{\sqrt{\frac{D_{\alpha }}{1000}}},$$
where: %Mn, %Si, %P, %Cu, %N, %Ni, %C—content in mass percent, respectively: manganese, silicon, phosphorus, copper, nitrogen, nickel, carbon in the steel, Dα—ferrite grain size, μm.

Figure 9 presents examples of the microstructure of the studied steel for several variants of controlled cooling after the deformation process.

The graph of changes in selected mechanical properties of 20MnB4 steel and average ferrite grain size depending on the cooling rate after the deformation process is shown in Figure 10.

Analysing the results of the metallographic studies of 20MnB4 steel after deformation and controlled cooling according to variant W1-1 (Figure 9a), it can be concluded that the examined steel in its entire volume had a banded microstructure in the form of alternately arranged bands of ferrite and perlite. In addition, some heterogeneity in the size and shape of the ferrite grain was observed. Based on the observations of microstructures obtained after the deformation and cooling process according to variant W1-2 (Figure 9b), it can be seen that the banding in the tested steel decreased and it occurred only in some areas of the tested samples. Nevertheless, the microstructure of the tested steel was still characterized by heterogeneous ferrite grain size. After the deformation of 20MnB4 steel and cooling at rates from about 3 °C/s to 15 °C/s (Figure 9c,d), no microstructure banding was observed in the tested material. In addition, greater uniformity of the microstructure was found. Only when cooling was at 15 °C/s did the ferrite grains in the 20MnB4 steel begin to take on a somewhat irregular shape. The research shows that there is an effect of the cooling rate after deformation on the microstructural structure of the investigated steel. It was found that in the studied cooling rate range, 20MnB4 steel had a ferritic–pearlitic microstructure. By choosing the right cooling rate for the material after deformation, a homogeneous fine-grained microstructure without clear banding can be obtained, which will ensure the required mechanical and technological properties are achieved.

Based on the data presented in Figure 10, it was found that in the studied cooling rate range after the deformation process there was a simultaneous increase in the yield strength and tensile strength of the tested steel grade together with an increase in the cooling rate. After cooling 20MnB4 steel at 0.6 °C/s, the yield strength was 322 MPa and the ultimate tensile strength was 508 MPa. After cooling the tested material at the rate of 15°C/s, the value of the yield strength increased to 433 MPa, while the ultimate tensile strength increased to 570 MPa. With an increasing cooling rate, the plasticity reserve of the tested steel (YS/UTS) also increased in the range from 0.63 to 0.76. Comparing the values of the analysed mechanical properties obtained after numerical and physical modelling (Figure 7 and Figure 10), they were found to be highly convergent. In the case of the offset yield strength, the largest error was about 8%, while in the case of tensile strength, less than 7%. Analysing the course of changes in ferrite grain size (Figure 10), it was found that this size decreased along with an increase in the cooling rate after deformation in the range from 19 μm to 6 μm.

### 3.3. Industrial Verification of Numerical and Physical Modelling of Wire Rod Rolling Process

The verification of the numerical and physical modelling of the rolling process of the wire rod was carried out for two variants differing in the cooling rate in the STELMOR^®^ line (W1-4 and W1-5). As part of the industrial verification, among others, temperature measurements of the 20MnB4 steel were carried out at several places in the rolling mill (including before and after the first rolling stand of the continuous bar rolling mill, before and after the cooling zones located between the continuous mill and the NTM block, between the NTM and RSM blocks, after the RSM block and STELMOR^®^ line). Figure 11 shows examples of thermograms for temperature distribution during industrial verification of the research.

As is apparent from the data presented in Figure 11, the average surface temperature of the 20MnB4 steel charge before the rolling process was about 1055 °C, while the average surface temperature of the wire rod at the beginning of the roller conveyor in the STELMOR^®^ system was about 860 °C. 

After comparing the measured temperature values with the numerically calculated values (Chapter 3.1), they were found to be highly compliant (maximum error of 7%). On this basis, it was found that the rheological properties of the studied steel, friction conditions and heat transfer coefficients adopted for numerical modelling of the process were correctly determined.

The next stage of industrial verification was metallographic testing of the received wire rod. Sample micrographs of the 20MnB4 steel microstructure in the longitudinal and cross-section are shown in Figure 12.

Based on the micrographs of the microstructure of the wire rod obtained in industrial conditions, cooled after rolling at the rate of 5°C/s (Figure 11a,b), it can be stated that the obtained product has a homogeneous ferrite grain size of a ferritic–pearlitic structure with small banding. In this case, the average ferrite grain size was about 9 μm, while in the longitudinal section it was about 10 μm. Comparing the average ferrite grain size in the wire rod obtained in industrial conditions with the average grain size obtained as a result of physical modelling (Chapter 3.2), a high consistency of the ferrite grain size was found. The error between the ferrite grain size measured on the wire rod cross-section and obtained during physical modelling was 8%. Nonetheless, the error between the ferrite grain size measured on the longitudinal section of the received wire rod and obtained as a result of physical modelling was 3%.

Analysing the results of the metallographic studies of 5.5 mm diameter wire rod cooled at the rate of 10°C/s (Figure 12c,d), it was observed that by increasing the cooling rate after rolling, much less banding of the 20MnB4 steel was obtained. Favourable microstructure fragmentation and even greater homogeneity in terms of ferrite grain size were also found. The average ferrite grain size on the cross-section and longitudinal section of the so manufactured wire rod was about 8 μm. The error between the ferrite grain size measured on the wire rod cross-section and that obtained as a result of physical modelling was just over 9%. In turn, the error between the ferrite grain size measured on the longitudinal section of the wire rod and that obtained as a result of physical modelling for this technological variant was below 1.5%. On this basis, it can be concluded that the average ferrite grain size obtained in industrial conditions is similar to the grain size obtained by the samples after the physical modelling of the rolling process according to the analysed variant.

The penultimate stage of industrial verification was testing selected mechanical and technological properties of the wire rod. Examples of tensile curves (in accordance with PN-EN ISO 6892-1:2016-09 [61]) for the verified variants are shown in Figure 13, while the exact values of the analysed mechanical and technological properties are given in Table 9. This table also includes the values of the total angle of non-dilatational strain and the total longitudinal true strain determined from the relationships proposed in paper [62]. The data presented in Table 9 are mean values of several tests carried out for each variant.

Based on the results of the measurements of the mechanical properties of the wire rod obtained after cooling on a roller conveyor according to variants W1-4 and W1-5, it can be concluded that the roller wire cooled on a roller conveyor at the rate of 10°C/s (variant W1-5) had a better complex of mechanical properties. This wire had higher values for YS by 8% and for UTS by over 6%. At the same time, there was a 10% decrease in elongation for this wire rod; however, this did not adversely affect the deformability of the examined wire rod, which was confirmed by the results of the upset tests. The narrowing of the wire rod for both variants was similar. After cooling the wire rod on a roller conveyor at the rate of 10 °C/s, a favourable increase in the plasticity reserve (YS/UTS) of about 1% was also found. The wire rod cooled on a roller conveyor at the rate of 10 °C/s was also characterized by higher values for the technological properties. The finished product obtained in these conditions also had higher values of the total angle of non-dilatational strain and the total longitudinal true strain.

The error between the (average) YS and UTS values determined in the static tensile test and the theoretically calculated (Chapter 3.1) and analytical dependencies (Chapter 3.2) values did not exceed 9%.

The final stage of industrial research was to determine the capacity of the obtained 20MnB4 steel wire rod for further cold forming. To achieve this aim, studies were carried out in upset tests in accordance with PN-83/H-04411 [63]) and assessment of the surface quality for possible cracks. An exemplary view of samples after the upsetting process is shown in Figure 14.

There were no fractures, cracks or other surface defects on the surface of the upset samples, even after applying a relative plastic strain of 75% (sample height index after upsetting 0.25). 

Based on a comparison of the results received from the values obtained by the numerical and physical modelling of the analysed rolling mill process, it can be said that high compliance was achieved.

## 4. Discussion

The speed of implementing the results of theoretical calculations and tests on a laboratory scale in industrial conditions determines the development and dissemination of new technologies. Industrial research is the last but usually very expensive element of the implementation process. The costs of implementing new technologies can be significantly reduced by using modern numerical and physical modelling methods. Using the abovementioned methods, the conditions for the thermoplastic processing of 20MnB4 steel wire rods were determined, guaranteeing the receipt of a finished product with properties far exceeding the minimum requirements of currently applicable standards, which are similar to the properties of products offered by leading global manufacturers [38]. Based on the research carried out, the following conclusions were formulated:-the best cooling variant is the W1-5 variant, in which the cooling rate was 10 °C/s—such parameters of thermoplastic processing ensure that a final product with a favourable complex of mechanical and technological properties as well as a fine-grained, even microstructure, lacking clear banding is obtained,-the wire rod produced in this way has a high yield strength of 0.74 and can be cold deformed with a relative plastic strain of 75%, without compromising the consistency of the material,-cooling of the examined steel grade after rolling in the RSM block at the temperature of 850 °C and subsequent controlled cooling in the range of 0.6–15°C/s ensures that a ferritic–pearlitic microstructure in the wire rod is obtained, -in the examined range, an increase in the cooling rate causes an increase in the analysed mechanical and technological properties of wire rods from 20MnB4 steel,-in the studied cooling rate range, an increase in the cooling rate caused a simultaneous increase in the yield strength, tensile strength and yield strength of the investigated steel,-the results obtained during the industrial verification correspond with high accuracy to the results obtained from the numerical and physical modelling of the analysed rolling mill process. This confirms the correct definition of the initial and boundary conditions during numerical modelling, especially the rheological properties of the tested steel, friction conditions and heat transfer coefficients. 

## Figures and Tables

**Figure 1 materials-13-00711-f001:**
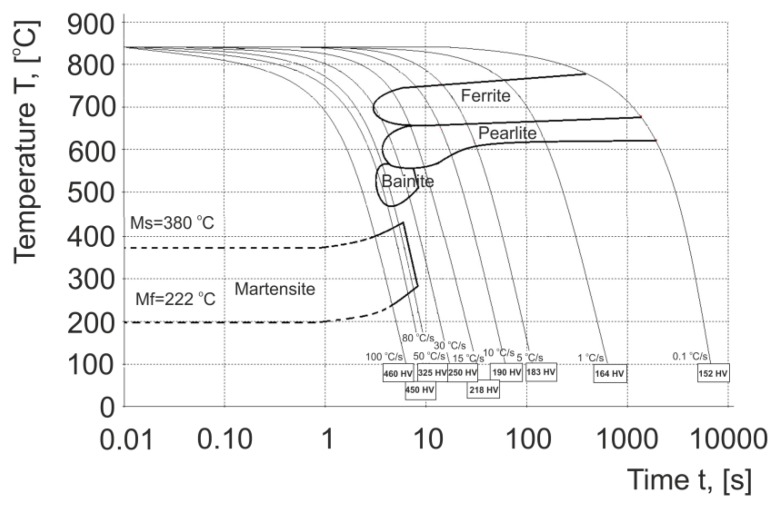
Real DTTT diagram for 20MnB4 steel [55]. Reproduced with permission from Laber, K., Koczurkiewicz, B., Determination of optimum conditions for the process of controlled cooling of rolled products with diameter 16.5 mm made of 20MnB4 steel, Proceedings of the 24th International Conference on Metallurgy and Materials—METAL 2015; published by Tanger Ltd., 2015.

**Figure 2 materials-13-00711-f002:**
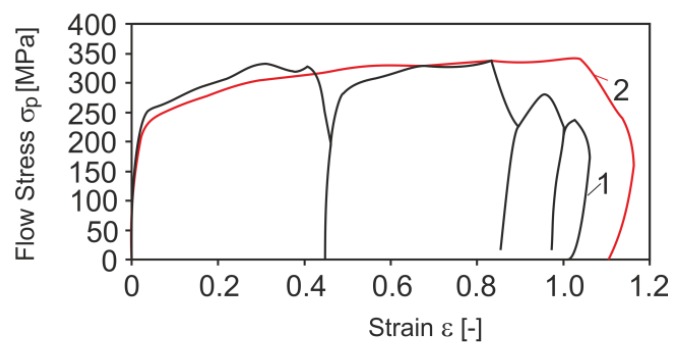
Change in yield stress of 20MnB4 steel during physical modelling of 16.5 mm diameter wire rod rolling, strip temperature in REDUCED SIZING MILL (RSM) block 860°C [58]: (**1**) sequence of four deformations; (**2**) single deformation. Reproduced with permission from Laber, K., Dyja, H., Koczurkiewicz, B., Sawicki S., Physical modeling of the wire rod rolling process of 20MnB4 steel, Proceedings of the VI Scientific Conference Rolling Mill Practice 2014. Processes-Tools-Materials; published by Akapit, 2014.

**Figure 3 materials-13-00711-f003:**
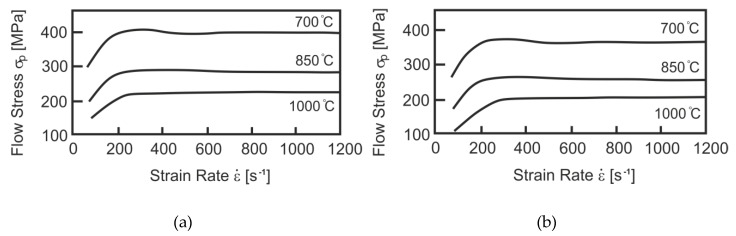
Impact of strain rate and temperature on yield stress of C35 steel: (**a**) at strain value 30%; (**b**) at strain value 17%.

**Figure 4 materials-13-00711-f004:**
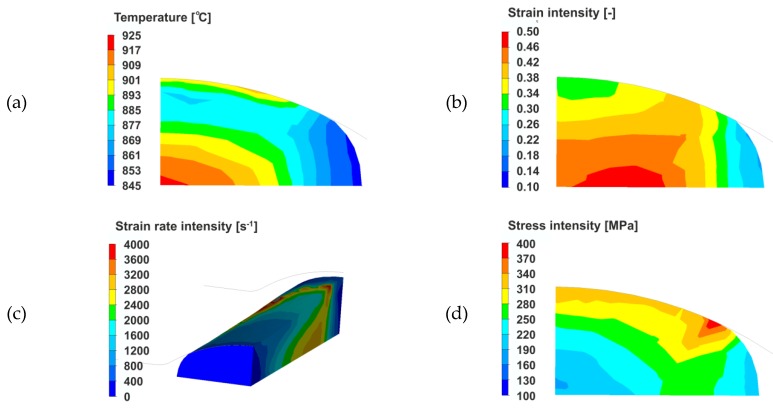
Sample results of numerical modelling of wire rod rolling process for rolling stand No. 1 in the RSM rolling mill block (rolling pass No. 28 in whole rolling period): (**a**) temperature distribution in strip exit plane from the strain zone; (**b**) strain intensity distribution in strip exit plane from strain zone; (**c**) strain rate intensity distribution; (**d**) stress intensity distribution in strip exit plane from strain zone.

**Figure 5 materials-13-00711-f005:**
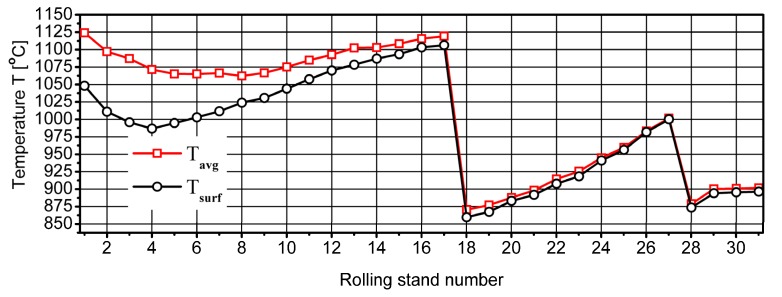
Change in 20MnB4 steel grade temperature (after subsequent passes) during 5.5 mm diameter wire rod rolling, in all rolling stands.

**Figure 6 materials-13-00711-f006:**
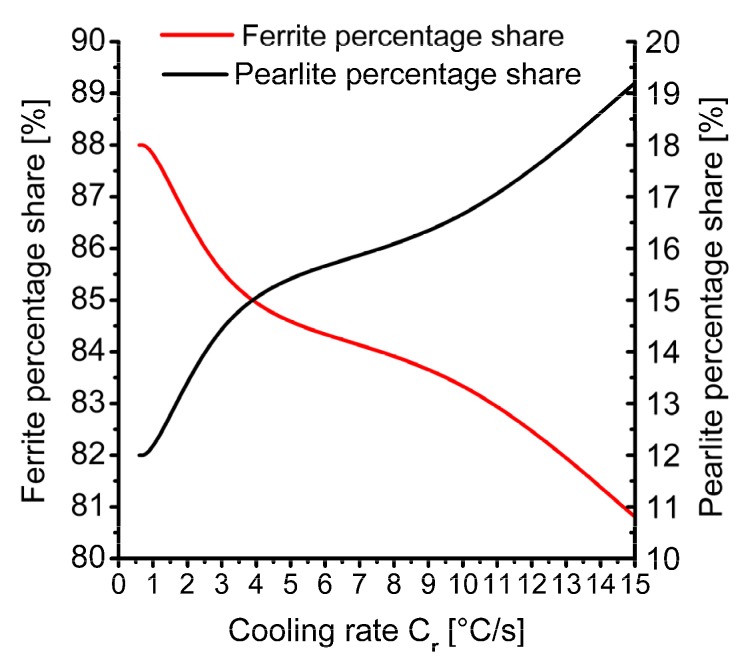
Influence of cooling rate after rolling process on percentage share of microstructure components on 5.5 mm diameter wire rod cross-section made of 20MnB4 steel.

**Figure 7 materials-13-00711-f007:**
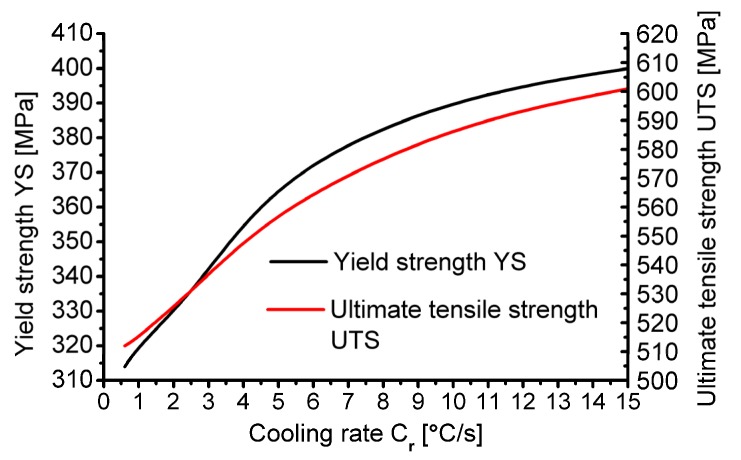
Influence of cooling rate of 5.5 mm diameter wire rod after rolling process on hardness and mechanical properties.

**Figure 8 materials-13-00711-f008:**
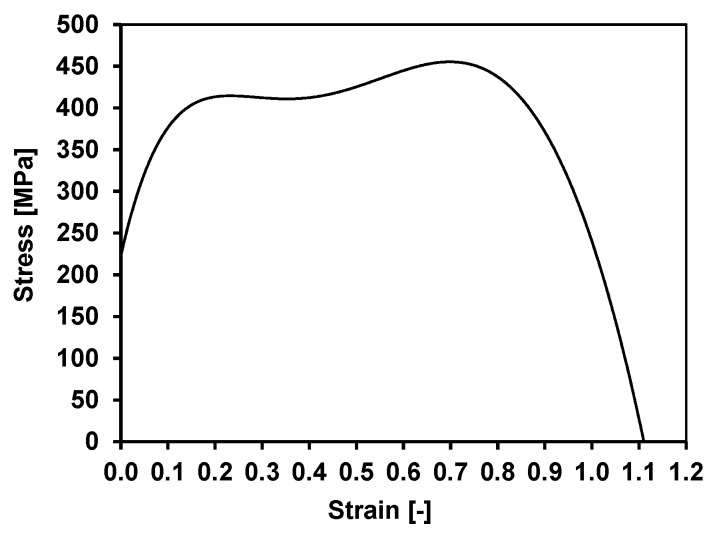
Change in 20MnB4 steel grade stress during physical modelling of 5.5 mm diameter wire rod rolling process.

**Figure 9 materials-13-00711-f009:**
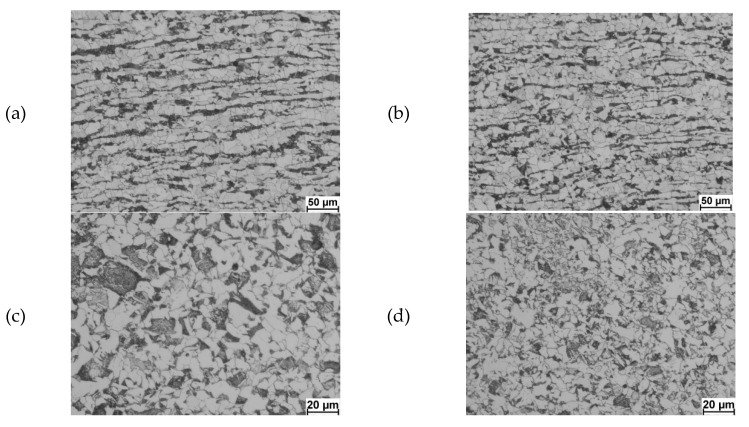
20MnB4 steel microstructure after physical modelling of 5.5 mm diameter wire rod rolling process: (**a**) cooling method W1-1, magnification 200×; (**b**) cooling method W1-2, magnification 200×; (**c**) cooling method W1-4, magnification 500×; (**d**) cooling method W1-5, magnification 500×.

**Figure 10 materials-13-00711-f010:**
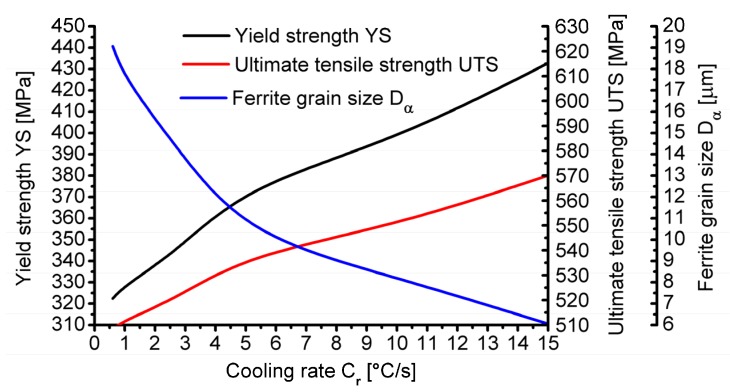
Influence of cooling rate after physical modelling of 5.5 mm diameter wire rod rolling process on mechanical properties and ferrite grain size of 20MnB4 steel.

**Figure 11 materials-13-00711-f011:**
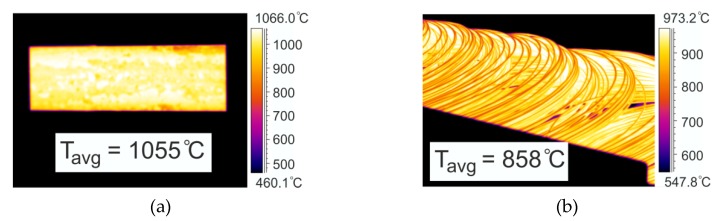
Thermogram examples of temperature distribution on wire rod surface: (**a**) before rolling mill stand No. 1; (**b**) at entry to roller conveyor of STELMOR^®^ line.

**Figure 12 materials-13-00711-f012:**
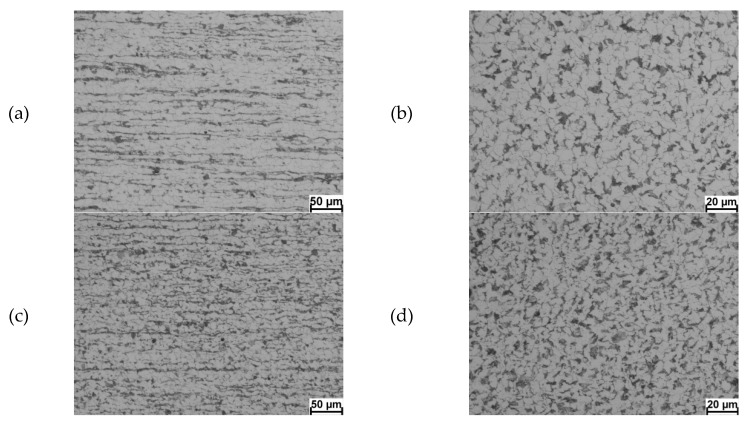
Microstructure of 5.5 mm diameter wire rod of 20MnB4 steel after rolling process in industrial conditions: (**a**,**b**) cooling method W1-4; (**c**,**d**) cooling method W1-5; (**a**,**c**) longitudinal section, magnification 200×; (**b**,**d**) cross-section magnification 500×.

**Figure 13 materials-13-00711-f013:**
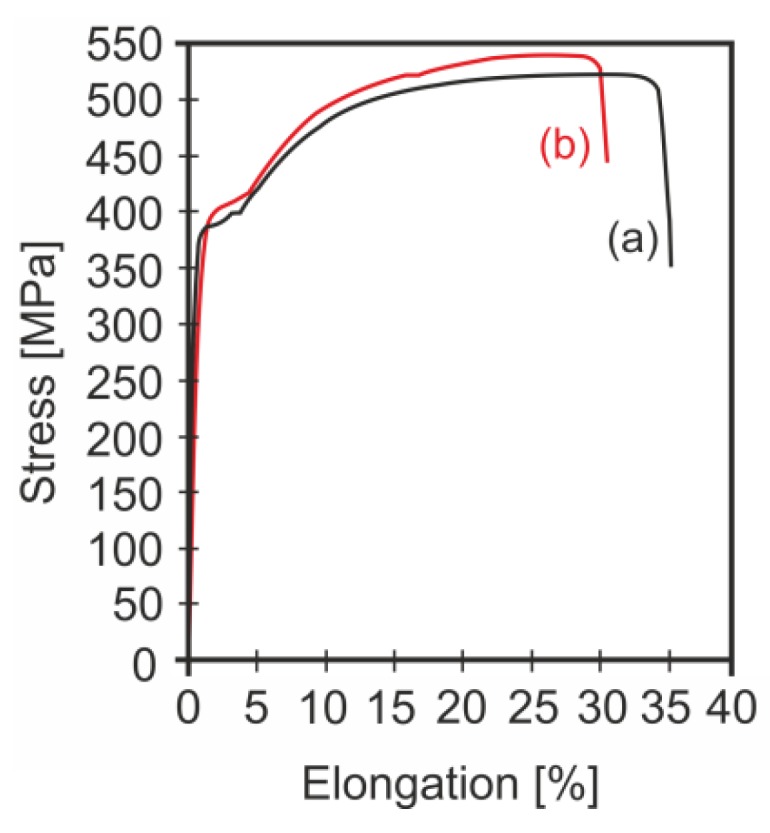
Examples of tensile curves of 5.5 mm diameter wire rod made of 20MnB4 steel: (**a**) cooling method W1-4; (**b**) cooling method W 1-5.

**Figure 14 materials-13-00711-f014:**
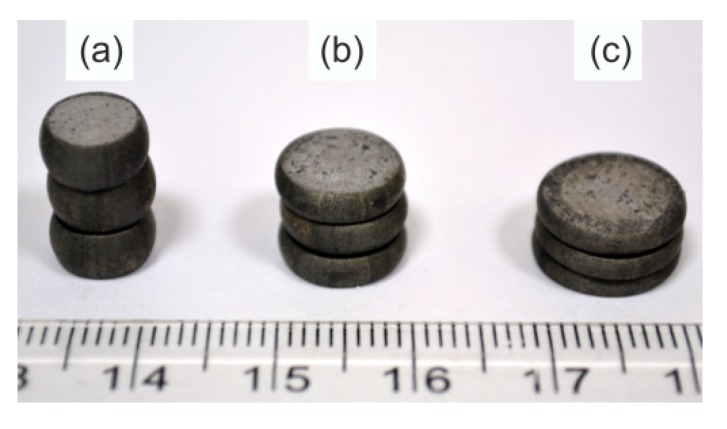
View of wire rod manufactured in accordance with W1-5 variant after upsetting process with relative plastic strain: (**a**) 50%; (**b**) 67%; (**c**) 75%.

**Table 1 materials-13-00711-t001:** Chemical composition of 20MnB4 steel [36].

Steel Grade	Steel Number	Melt Analysis, mass%						
20MnB4	1.5525	C	Si	Mn	P_max_, S_max_	Cr	Cu_max_	B
		0.18–0.23	≤0.30	0.90–1.20	0.025	≤0.30	0.25	0.0008–0.005

**Table 2 materials-13-00711-t002:** Parameters of heat treatment after rolling 20MnB4 steel rod with diameter of 5.5 mm ^1^.

Surface Temperature before RSM Block 850 °C		1 Cooling Stage ^2^	
		Desired Surface Temperature T_surf_, °C	Cooling Rate C_r_, °C/s
**Cooling variant in STELMOR^®^ line (Primetals Technologies USA LLC, Alpharetta, GA, USA)**	W1-1	575	0.6
	W1-2	500	1
	W1-3	500	3
	W1-4	500	5
	W1-5	500	10
	W1-6	500	15

^1^ Table based on data published in work [3]; ^2^ In second cooling stage, studied steel was cooled to 200 °C at a rate of 1 °C/s.

**Table 3 materials-13-00711-t003:** Boundary conditions for numerical modelling of rolling process of 5.5 mm wire rod ^1^.

Temperature			Strip Heat Transfer Coefficients From:			Friction Factor m,	Coefficient of Friction μ,
Air $T_{air}$,	Water $T_{water}$,	Rolls $T_{rolls}$,	Air $\alpha_{air}$,	Rolls $\alpha_{rolls}$,	Water $\alpha_{water}$,		
${}^{{}^{\circ}}C$	${}^{{}^{\circ}}C$	${}^{{}^{\circ}}C$	$W/m^{2}K$	$W/m^{2}K$	$W/m^{2}K$	-	-
20	20	60	100	3000–5000	6700–16000	0.56–0.8	0.28–0.4

^1^ Table based on data published in work [3].

**Table 4 materials-13-00711-t004:** Thermophysical properties of 20MnB4 steel ^1^.

Thermophysical Properties of Steel		
Thermal Conductivity λ, W/mK	Density ρ, $kg/m^{3}$	Specific Heat $c_{p}$, J/kgK
35.5	7850	778

^1^ Data based on material base of FORGE 2011^®^ program.

**Table 5 materials-13-00711-t005:** Equation coefficients (6) used during numerical modelling of 5.5 mm wire rod rolling in NTM and RSM blocks of rolling mill [3].

A	m_1_	m_2_	m_3_	m_4_	m_5_	m_7_	m_8_	m_9_
707,153 × 10^7^	0.0012	0.1943	0.0424	−0.0031	−0.0004	−0.0721	0.00002	−3.7326

**Table 6 materials-13-00711-t006:** Characteristic temperatures of phase transitions of 20MnB4 steel ^1^.

Cooling Rates C_r_ [°C/s]	Characteristic Temperatures [°C]
100	M_s_ = 406, M_f_ = 248
80	B_s_ = 550, B_f_ = 460, M_s_ = 420, M_f_ =3 70
50	B_s_ = 560, B_f_ = 450, M_s_ = 430, M_f_ = 324
30	F_s_ = 700, F_f_ = 680, P_s_ = 630, P_f_ = B_s_ = 550, B_f_ = 485
15	F_s_ = 729, F_f_ = P_s_ = 650, P_f_ = 560
10	F_s_ = 757, F_f_ = P_s_ = 670, P_f_ = 562
5	F_s_ = 743, F_f_ = P_s_ = 670, P_f_ = 618
1	F_s_ = 760, F_f_ = P_s_ = 660, P_f_ = 633
0.1	F_s_ = 790, F_f_ = P_s_ = 692, P_f_ = 632
where: Ps, Pf, Fs, Ff, Bs, Bf, Ms, Mf—start and end temperature of phase transitions, respectively: perlitic, ferritic, bainitic, martensitic	

^1^ Table based on data published in work [55]. Reproduced with permission from Laber, K., Koczurkiewicz, B., Determination of optimum conditions for the process of controlled cooling of rolled products with diameter 16.5 mm made of 20MnB4 steel, Proceedings of the 24th International Conference on Metallurgy and Materials—METAL 2015; published by Tanger Ltd., 2015.

**Table 7 materials-13-00711-t007:** Parameters of wire rod rolling process of 20MnB4 steel grade ^1^.

Pass Number	Strain Intensity $\varepsilon_{i}$ [-]	Strain Rate Intensity ${\dot{\varepsilon }}_{i}$ [s^−1^]	Stress Intensity $\sigma_{i}$ [MPa]
Continuous rolling mill			
1	0.18	0.16	74.76
2	0.39	0.35	96.02
3	0.28	0.39	92.81
4	0.59	0.96	109.57
5	0.46	1.15	103.10
6	0.50	2.02	115.55
7	0.45	2.45	117.61
8	0.48	4.71	123.21
9	0.44	5.57	130.59
10	0.54	10.39	138.72
11	0.48	12.07	134.07
12	0.50	20.53	142.58
13	0.51	24.74	143.39
14	0.50	46.34	152.09
15	0.41	47.13	148.10
16	0.51	79.93	154.04
17	0.31	70.63	139.36
NTM block of wire rod rolling mill			
18	0.32	156.02	229.99
19	0.51	171.25	211.09
20	0.56	276.33	225.24
21	0.54	303.93	213.58
22	0.56	477.46	224.07
23	0.53	584.28	215.56
24	0.62	991.51	239.33
25	0.57	1042.10	198.68
26	0.62	1753.46	232.37
27	0.41	1809.67	181.90
RSM block of wire rod rolling mill			
28	0.53	2368.05	399.89
29	0.48	2275.43	385.39
30	0.13	1853.11	378.54
31	0.10	1680.68	374.72

^1^ Table based on data published at in work [3]. Reproduced with permission from Laber, K., New Aspects of Wire Rod Production from Steel for Cold Heading, Series: Monograph No. 79; published by Czestochowa University of Technology, Faculty of Production Engineering and Materials Technology Publishing House, 2018

**Table 8 materials-13-00711-t008:** Changes in 20MnB4 steel austenite grain size (Dγ) during 5.5 mm diameter wire rod rolling process.

**Pass No.**	**Continuous Rolling Mill**																
	**1**	**2**	**3**	**4**	**5**	**6**	**7**	**8**	**9**	**10**	**11**	**12**	**13**	**14**	**15**	**16**	**17**
D_γ_ [μm]	215	120	127	88	84	76	84	67	65	62	70	68	73	64	63	63	44
**Pass No.**	**NTM Rolling Mill Block**											**RSM Rolling Mill Block**					
	**18**		**19**	**20**	**21**	**22**	**23**	**24**	**25**	**26**	**27**	**28**		**29**	**30**		**31**
D_γ_ [μm]	43		17	17	17	18	17	18	19	20	42	21		19	18		18

**Table 9 materials-13-00711-t009:** Mechanical and technological properties of 5.5 mm diameter wire rod made of 20MnB4 steel after rolling process.

Cooling Variant	Yield Strength YS [MPa]	Ultimate Tensile Strength UTS [MPa]	Unit Elongation A_5_ [%]	Relative Reduction of Area at Fracture Z [%]	Number of Twists to Break N_t_	Number of Bends to Break N_b_	Total Angle of Non-Dilatational Strain γ [°]	Total Longitudinal True Strain ε_l_
W1-4	386	525	33.1	69.4	38.1	26.2	67.3	0.95
W1-5	415	559	29.8	69.7	41.3	29.6	68.9	1.02

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
