# Peer review of "Determining Conditions for Thermoplastic Processing Guaranteeing Receipt of High-Quality Wire Rod for Cold Upsetting Using Numerical and Physical Modelling Methods"

_materials, 2020, doi:10.3390/ma13030711_

Round 1

Reviewer 1 Report

The manuscript examines the thermoplastic processing of steel 20MnB4 wire rod. Mathematical and physical modeling of the process is considered. In general, this work is of undoubted theoretical and practical interest, however, before publication, appropriate additions and explanations should be made:

The abstract is too long and contains a lot of unnecessary information. I recommend that the authors reduce the abstract to a maximum of 200 words. Especially a lot of space is given to the description of research methods. But in the abstract, it’s enough to report very briefly about the main methods, transferring the details to the Materials and Methods section.

As a rule, the notation [23-30] is used, not [23 ÷ 30].

In mathematical modeling of high-temperature processes, taking into account the temperature effect is of particular importance. It is important to remember that when the temperature rises to sufficiently large values (500 degrees Celsius and above), the characteristics of the material begin to change nonlinearly, obeying complex laws, since it is necessary to take into account phase transformations in the material. The friction coefficient also changes according to a complex nonlinear law (first increasing, and then, as a rule, decreasing), moreover, this pattern will be individual for each material. The change in the coefficient of friction is significantly affected by the formation of oxide films on the surface of metals and compounds. These processes cannot be taken into account due to the introduction of various correction factors, since there are rather sharp transitions (for example, phase transitions), leading to a sharp and often significant change in the properties of the material, that is, at a temperature of, for example, 700 degrees, you are dealing with something completely different material with completely different properties. There it is more relevant for a temperature of 1130 degrees!

Those formulas used by the authors of (2) - (6) suggest a certain systematic change in the properties of the material depending on the temperature, but this is true only for a certain temperature range (until phase changes or active oxide formation on the surface or, for example, diffusion processes, etc.).

The influence of phase transitions is taken into account in formulas (8) - (11), but it is not very clear where the authors obtained data on the ratio of different phases in a specific material (20MnB4 steel).

How is the effect of oxide films on the friction coefficient taken into account?

The format of Table 3 should be fixed.

Line 188 - need to fix "work 3]"

Lines 261-62. I think that "is not possible" is not entirely correct - it is more correctly "impossible when using existing equipment."

Author Response

Response to Reviewer 1 Comments

Point 1: The abstract is too long and contains a lot of unnecessary information. I recommend that the authors reduce the abstract to a maximum of 200 words. Especially a lot of space is given to the description of research methods. But in the abstract, it’s enough to report very briefly about the main methods, transferring the details to the Materials and Methods section.

Response 1: Actually the abstract is too extensive. According to the reviewer suggestion, it was shortened to about 200 words. Some information from abstract is in "Materials and Methods" and "Results" chapters.

Point 2: As a rule, the notation [23-30] is used, not [23 ÷ 30].

Response 2: According to the reviewer suggestion We correct this notations in all manuscript.

Point 3: In mathematical modeling of high-temperature processes, taking into account the temperature effect is of particular importance. It is important to remember that when the temperature rises to sufficiently large values (500 degrees Celsius and above), the characteristics of the material begin to change nonlinearly, obeying complex laws, since it is necessary to take into account phase transformations in the material. The friction coefficient also changes according to a complex nonlinear law (first increasing, and then, as a rule, decreasing), moreover, this pattern will be individual for each material. The change in the coefficient of friction is significantly affected by the formation of oxide films on the surface of metals and compounds. These processes cannot be taken into account due to the introduction of various correction factors, since there are rather sharp transitions (for example, phase transitions), leading to a sharp and often significant change in the properties of the material, that is, at a temperature of, for example, 700 degrees, you are dealing with something completely different material with completely different properties. There it is more relevant for a temperature of 1130 degrees!

Those formulas used by the authors of (2) - (6) suggest a certain systematic change in the properties of the material depending on the temperature, but this is true only for a certain temperature range (until phase changes or active oxide formation on the surface or, for example, diffusion processes, etc.).

Response 3: Authors agree with Reviewer opinion, that the friction coefficient is not constant value in hot rolling process. One of the most important parameter which has influence on the friction coefficient is temperature. The others is strain rate, strain value, surface conditions of the material and rolls e.t.c. During numerical modelling of the rolling process we used combined models of friction (Tresca and Coulomb models). We use friction coefficient and friction factor as depending on the temperature, based on Our earlier experience (for technological conditions of this rolling mill department), where we compared for example shape of finished product or energy and force parameters - obtained by numerical modeling - with industrial results, among other things (list of some papers below). Based on these results we found that accuracy of calculated shape (dimensions) and energy and force parameters is close to these measured in real rolling process in industrial conditions. So we think that values of friction coefficient and friction factor are corrected, because friction has influence on shape (dimensions) and energy and force parameters, among other thing. Moreover the friction force - as we can see from equation (2) and (3) is depended on yield stress (flow stress) which is depended on temperature, strain and strain rate. The yield stress (flow stress) is not linear function. Naturally we did investigations of yield stress (flow stress) of 20MnB4 steel – rheological properties (table 4), for all temperature and strain conditions range during wire rod rolling process. These results were imported to the FORGE 2011 database. Changes of yield stress (flow stress) during hot rolling were described by Hensel-Spittel equation (equation no. 6), which consider influence of strain, strain rate and temperature on yield stress (flow stress) changes. This equation is often used (with good results) for numerical modelling of hot working processes. Unfortunately there is no results of energy and force parameters and dimension accuracy in this paper. We did it of course and the results is very good. The difference between measured and obtained by numerical modelling diameter is less than 2%. For the measured and calculated energy and force parameters the difference was less than 10%.

Equations no. 4 is Fourier equation for the undetermined heat flow which is proper for hot rolling process. Eguation no. 5 is the boundary conditions equation (combined boundary conditions of the second and third types) – proper for heat exchange during rolling process in Our opinion.

List of some papers where we compared for example shape of finished product or energy and force parameters - obtained by numerical modeling - with industrial results, among other things:

Laber K., Mróz S.: „WpÅ‚yw walcowania normalizujÄ…cego na dokÅ‚adność wymiarowÄ… prÄ™tów okrÄ…gÅ‚ych”, materiaÅ‚y konferencyjne IX MiÄ™dzynarodowej Konferencji Naukowej pt.: „Nowe technologie i osiÄ…gniÄ™cia w metalurgii i inżynierii materiaÅ‚owej”, pr. zbior. pod red. nauk. Lecha Szecówki, CzÄ™stochowa 30 maja 2008 r., Wydawnictwo Politechniki CzÄ™stochowskiej, CzÄ™stochowa 2008 r., ISBN 978-83-7193-372-1, s. 238÷241. Laber K.: „Modelowanie i optymalizacja procesów regulowanego walcowania i kontrolowanego chÅ‚odzenia wyrobów walcowni bruzdowych”, praca doktorska Politechnika CzÄ™stochowska WydziaÅ‚ Inżynierii Procesowej, MateriaÅ‚owej i Fizyki Stosowanej, CzÄ™stochowa 2008. Mróz S., Dyja H., Laber K.: „Influence of the round bars normalizing rolling process on the energy and force parameters”, Steel Research International, Special Edition Metal Forming, Volume 1, 2008, Proceedings of the 12th International Conference „Metal Forming 2008”, Akademia Górniczo-Hutnicza (AGH University of Science and Technology), September 21.-24. 2008, Kraków, Poland, Edited by: Maciej Pietrzyk, Jan Kusiak, Janusz Majta, Peter Hartley, Jianguo Lin, Kenichiro Mori, Publishing Company Verlag Stahleisen GmbH, Düsseldorf, Germany 2008, ISBN 978-3-514-00754-3, ISSN 1611-3683, 410÷416. Laber K: „Modelowanie i optymalizacja procesów regulowanego walcowania i kontrolowanego chÅ‚odzenia wyrobów walcowni bruzdowych”, Monografia zbiorowa pod redakcjÄ… Henryka Dyi pt.: „Metalurgia 2009 – Nowe technologie i osiÄ…gniÄ™cia”, seria: Monografie nr 1, Politechnika CzÄ™stochowska, WydziaÅ‚ Inżynierii Procesowej, MateriaÅ‚owej i Fizyki Stosowanej, Wydawnictwo WydziaÅ‚u Inżynierii Procesowej, MateriaÅ‚owej i Fizyki Stosowanej Politechniki CzÄ™stochowskiej, ISBN 978-83-87745-13-4, ISSN 2080-2072, CzÄ™stochowa 2009r., s. 99-122. Laber K. B., Dyja H. S., Mróz S. J.: „The influence of rolling temperature on the energy and force parameters during normalizing rolling of plain round bars”, Materials Science Forum Vols. 638-642 (2010), ISSN 1662-9752, pp. 2628-2633. Laber K., Sygut P., Mróz S., Dyja H.: „Numerical verification of the rolls calibration for 200x6mm flat bars rolling”, Hutnik – WiadomoÅ›ci Hutnicze, rocznik R. 77, nr 5/2010, ISSN 1230-3534, s. 214-216. Sygut P., Laber K., Mróz S., Dyja H.: „WpÅ‚yw nierównomiernego rozkÅ‚adu temperatury na dÅ‚ugoÅ›ci wsadu na parametry energetyczno-siÅ‚owe podczas walcowania prÄ™tów okrÄ…gÅ‚ych”, Hutnik – WiadomoÅ›ci Hutnicze, rocznik R. 77, nr 9/2010, ISSN 1230-3534, s. 540-542. Laber K., Dyja H.: „Analiza parametrów energetyczno-siÅ‚owych podczas walcowania normalizujÄ…cego prÄ™tów okrÄ…gÅ‚ych gÅ‚adkich o Å›rednicy 38 mm w ciÄ…gÅ‚ej walcowni bruzdowej”, Archiwum Technologii Maszyn i Automatyzacji, vol. 30, nr 3, ISSN 1233-9709, 2010, s. 139-146. Sygut P., Mróz S., Laber K., Dyja H.: „Influence of the non-uniform temperature distribution on the metallic charge length on dimensional accuracy round bars”, 7th International Conference Mechatronic Systems and Materials (MSM 2011)
7 - 9 July, 2011, Kaunas, Lithuania, ISSN 1822-8283. Laber K., Sygut P., Mróz S., Dyja H.: „ Application of thermovision technique for the investigation of influence of temperature change on the energy and force parameters during industrial plain round bars rolling process”, 7th International Conference Mechatronic Systems and Materials (MSM 2011), 7 - 9 July, 2011, Kaunas, Lithuania, ISSN 1822-8283. Sygut P., Laber K., Borkowski S.: „Investigation of the non-uniform temperature distribution on the metallic charge length during round bars rolling process”, Manufacturing Technology Vol.12 nr 13, ISSN 1213-2489, pp. 260-263. Laber K., Mróz S., Sygut P., Dyja H.: Analysis of the temperature change over the continuous ingot length on the parameters of round bar rolling process. Metalurgija, vol. 52, 1/2013, pp. 39÷42. Mróz S., Laber K., Sygut P., Dyja H.: Effect of temperature distribution over the feedstock length on the metal plastic flow during rod rolling. Steel Research International, Special Edition, 14th International Conference on Metal Forming 2012, 16÷19.09.2012, Kraków, Poland, pp. 119÷122.

Value of friction coefficient and friction factor which we used in numerical modelling – as depending on rolling conditions (temperature, strain rate, strain value):

Pass no.

Coefficient of friction

Friction factor

1÷13

0.4

0.8

14

0.32

0.64

15

0.4

0.8

16

0.32

0.64

17

0.28

0.56

18÷31

0.3

0.6

Point 4: The influence of phase transitions is taken into account in formulas (8) - (11), but it is not very clear where the authors obtained data on the ratio of different phases in a specific material (20MnB4 steel).

Response 4: The QTSteel software calculates the ratio of the various microstructural constituents and mechanical properties of steels after heat treatment (quenching, tempering). Beginning with the chemical composition and the austenitizing conditions of time and temperature the software calculates corresponding CCT diagram which describes the decomposition of the undercooled austenite to ferrite, pearlite, bainite and martensite.

Predicted or user defined CCT diagrams can be used for computer simulation of isothermal or anisothermal transformation of specified steel for predefined cooling curve. The software divides the cooling curve into the sequences with uniform cooling rate and calculates the percentage of the various microstructural constituents (ferrite, pearlite, bainite or martensite) for each sequence.

Detailed studies aimed at determining phase transition temperatures, methodology and elaboration of TTT and DTTT diagrams for 20MnB4 steel grade are presented and described in details in the paper no. [55]. There is an information about it in the manuscript. We added some results from the paper [55] to our manuscript (Characteristic temperatures of phase transitions and hardness of 20MnB4 steel – table 5, and DTTT diagram figure 1).

Point 5: How is the effect of oxide films on the friction coefficient taken into account?

Response 5: We didn’t investigate the effect of oxide films on the friction coefficient directly. Moreover (in industry conditions) – before first pass the band was cleaned by high pressure water system (Descaler) to remove the scale from the surface. Moreover the speed of rolling is high (about 100 m/s in last passes) and the time between passes is short. Additionally all rolling stands in NTM and RSM rolling blocks are "covered". Based on Our earlier investigations for this rolling mill department (paper list in response no. 3) we think that we correct define friction conditions. To define friction conditions we used inverse method. We based on comparision of dimension accuracy and energy and force parameters (calculated by finite element method and measured in industry). For this wire rod rolling process the difference between measured and obtained by numerical modelling diameter is less than 2%. For the measured and calculated energy and force parameters the difference was less than 10%.

Value of friction coefficient and friction factor which we used in numerical modelling – as in response no. 3.

Point 6: The format of Table 3 should be fixed.

Response 6: According to the  Reviewer 1 suggestion format of Table 3 was changed.

Point 7: Line 188 - need to fix "work 3]"

Response 7: According to the  Reviewer 1 suggestion this mistake is corrected.

Point 8: Lines 261-62. I think that "is not possible" is not entirely correct - it is more correctly "impossible when using existing equipment."

Response 8: According to the  Reviewer 1 suggestion this sentence was changed: "Accurate physical modelling of the four-deformation cycle while maintaining the appropriate break times between successive passes is impossible when using existing equipment."

Additionally:

References was formatted according to the „Materials” Journal reqirements and „DOI” identifier were added in some places also. We added also translations in works other than in English language. All manuscript was verified in grammar by professional translator.

Reviewer 2 Report

It is an interesting work focusing on the determining conditions for thermoplastic processing guaranteeing receipt of high-quality wire rod for cold upsetting using numerical and physical modelling methods.

It is a nice piece of work, following comments are needed considered:

1. The introduction part, even if the authors make an extensive literature survey, for the reviewer, it is not clear enough what is the novel for the work, the authors need to clarify this. 

2. For the Figure 5 of fraction the microstructure, what can be the error bar range for each?

3. Equations (12) and (13) express the empirical equations to calculate YS and UTS, an open question is how the accuracy compare with the state of the art models, e.g. statistical models, etc., this is not an mandatory comment.  

4. The conclusions can be much shorter and clearer for stating a few points.

Author Response

Response to Reviewer 2 Comments

Point 1: The introduction part, even if the authors make an extensive literature survey, for the reviewer, it is not clear enough what is the novel for the work, the authors need to clarify this.

Response 1: According to the Reviewer 2 suggestion the last paragraph of the manuscript introduction was modified : „In the available technical literature, there are few papers on the rolling process of wire rod which describe the possibilities of shaping and improving the mechanical and technological properties of the finished product using numerical methods and physical modelling taking into account the limitations of the available testing apparatus and the verification of such studies in industrial conditions. Therefore, the research issues undertaken at work are current. An important achievement of the work is solution of the numerical and physical modeling problems of the analyzed rolling process using commercially available software and test equipment, taking into account its limitations in terms of the applied total strain, strain rate and break times between successive deformations. The proposed methodology for modelling the rolling process of the wire rod reflects with high accuracy the actual technological process and changes occurring in the microstructure of the deformed material. The proposed parameters of thermo-plastic processing of wire rod from 20MnB4 steel grade with diameter of 5.5 mm ensure that a finished product with a microstructure and properties comparable with the products offered by leading world producers is obtained. The obtained results and their analysis should be helpful in developing changes in the currently used methods of wire rod production, or in the design of new technological lines for rolling wire rod.”

Point 2: For the Figure 5 of fraction the microstructure, what can be the error bar range for each?

Response 2: The QTSteel software calculates the ratio of the various microstructural constituents and mechanical properties of steels after heat treatment (quenching, tempering). Beginning with the chemical composition and the austenitizing conditions of time and temperature the software calculates corresponding CCT diagram which describes the decomposition of the undercooled austenite to ferrite, pearlite, bainite and martensite.

Predicted or user defined CCT diagrams can be used for computer simulation of isothermal or anisothermal transformation of specified steel for predefined cooling curve. The software divides the cooling curve into the sequences with uniform cooling rate and calculates the percentage of the various microstructural constituents (ferrite, pearlite, bainite or martensite) for each sequence.

Detailed studies aimed at determining phase transition temperatures, methodology and elaboration of TTT and DTTT diagrams for 20MnB4 steel grade are presented and described in details in the paper no. [55]. Laber K.; Koczurkiewicz B. Determination of optimum conditions for the process of controlled cooling of rolled products with diameter 16.5 mm made of 20MnB4 steel. 24th International Conference on Metallurgy and Materials - METAL 2015, Brno, Czech Republic, June 3rd÷5th 2015, pp. 364÷370.

The accuracy of determining the percentage share of microstructure components depends on the DTTT diagram implementation accuracy into the QTSteel program database, among other things. It is difficult to say what can be the error bar range for each microstructure component because the share of microstructure components in the real material (wire rod) has not been investigated in the paper unfortunately. So, it is impossible to compare results in this field. But based on others results (mechanical and technological properties compared between real wire rod and material after numerical and physical simulation) it can be stated that the results are very consistent. So, in our opinion the maximum error bar range for each microstructure component can be less than  about 10%.

Point 3: Equations (12) and (13) express the empirical equations to calculate YS and UTS, an open question is how the accuracy compare with the state of the art models, e.g. statistical models, etc., this is not an mandatory comment. 

Response 3: These equations are taken from technical literature ([60]. Hodgson P.D.; Gibbs R.K. A mathematical model to predict the mechanical properties of hot rolled C-Mn and microalloyed steels. ISIJ International. 1992, vol. 32, 12, pp. 1329÷1338; https://doi.org/10.2355/isijinternational.32.1329). They are dedicated for C-Mn steels. Results obtained by using these equations were compared also with others empirical equations (Sawada Y., Foley R.P., Thompson S.W., Krauss G.: Proc. 35th MWSP Conf. Proc. ISS-AIME, Pitsburgh 1994, p. 263):

(1)

(2)

where: HV –Vickers hardness

and with compression test results (for YS) of samples after physical modeling.

Results obtained by all these methods are comparable (average differences about 7%). Unfortunately, the others art models, e.g. statistical models, etc, were not used.

Point 4: The conclusions can be much shorter and clearer for stating a few points.

Response 4: According to the Reviewer 2 suggestions, chapter no.4 was modified:

„The speed of implementing the results of theoretical calculations and tests on a laboratory scale in industrial conditions determines the development and dissemination of new technologies. Industrial research is the last but usually very expensive element of the implementation process. The costs of implementing new technologies can be significantly reduced by using modern numerical and physical modelling methods. Using the above-mentioned methods, the conditions for the thermoplastic processing of 20MnB4 steel wire rods were determined, guaranteeing the receipt of a finished product with properties far exceeding the minimum requirements of currently applicable standards, which are similar to the properties of products offered by leading global manufacturers [38]. Based on the research carried out, the following conclusions were formulated:

-   the best cooling variant is the W1-5 variant, in which the cooling rate was 10°C/s, such parameters of thermoplastic processing ensure a final product with a favourable complex of mechanical and technological properties as well as a fine-grained, even microstructure, lacking clear banding is obtained,

-   the wire rod produced in this way has a high yield strength of 0.74 and can be cold deformed with a relative plastic strain of 75%, without compromising the consistency of the material,

-   cooling of the examined steel grade after rolling in the RSM block at the temperature of 850°C and subsequent controlled cooling in the range of 0.6÷15°C/s ensures a ferritic-pearlitic microstructure in the wire rod is obtained,

-   in the examined range, an increase in the cooling rate causes an increase in the analysed mechanical and technological properties of wire rod from 20MnB4 steel,

-   in the studied cooling rate range, an increase in the cooling rate caused a simultaneous increase in the yield strength, tensile strength and yield strength of the investigated steel,

-   the results obtained during the industrial verification correspond with high accuracy to the results obtained from the numerical and physical modelling of the analysed rolling mill process. This confirms the correct definition of the initial and boundary conditions during numerical modelling, especially the rheological properties of the tested steel, friction conditions and heat transfer coefficients.”

Additionally:

References was formatted according to the „Materials” Journal reqirements and „DOI” identifier were added in some places also. We added also translations in works other than in English language. All manuscript was verified in grammar by professional translator.
